# A Fast Framework for Post-Training Structured Pruning without Retraining

## Abstract

Pruning has become a widely adopted technique for compressing and accelerating deep neural networks. However, most pruning approaches rely on lengthy retraining procedures to restore performance, rendering them impractical in many real-world settings where data privacy regulations or computational constraints prohibit extensive retraining. To address this limitation, we propose a novel framework for rapidly pruning pre-trained models without any retraining. Our framework focuses on structured pruning. It first groups coupled structures across layers based on their dependencies and comprehensively measures and marks the least important channels in a group. Then we introduce a two-phase layer reconstruction strategy utilizing a small amount of unlabeled data to recover the accuracy drop induced by pruning. The first phase imposes a sparsity penalty on less important channels to squeeze information into the remaining components before pruning. The second phase executes pruning and calibrates the layer output discrepancy between the pruned and original models to reconstruct the output signal. Experiments demonstrate that our framework achieves significant improvements over retraining-free methods and matches the accuracy of pruning approaches that require expensive retraining. With access to about 0.2% samples from the ImageNet training set, our method achieves up to 1.73x reduction in FLOPs, while maintaining 72.58% accuracy for ResNet-50. Notably, our framework prunes networks within a few minutes on a single GPU, which is orders of magnitude faster than retraining-based techniques.

## 1 Introduction

The recent emergence of edge computing applications calls for the necessity for deep neural compression. Main compression techniques include pruning (Han et al., 2015), quantization (Banner et al., 2018), knowledge distillation (Hinton et al., 2015), and low-rank decomposition (Jaderberg et al., 2014). Among these, pruning has proven widely effective for network compression and acceleration (Li et al., 2017; Lin et al., 2020; Lee et al., 2019; Dong et al., 2017). The different pruning methods can be roughly categorized into two schemes: unstructured pruning that zeros individual weights (Tanaka et al., 2020; Evci et al., 2020) and structured pruning that removes entire channels or blocks (Wang et al., 2021; He et al., 2019). Unstructured pruning produces sparse weight matrices and relies on special hardware to translate into faster execution. In contrast, structured pruning methods align well with hardware architectures, thereby enabling wider practical application.

Previous approaches involve a training process, regardless of whether pruning after training (Li et al., 2017), pruning during training (You et al., 2019; He et al., 2018), and pruning at initialization (Lee et al., 2019; Wang et al., 2020). The training process presents significant obstacles to their adoption in real-world settings. Due to privacy or commercial concerns, the original training data may be inaccessible or restricted for training pruned models. For example, commercial companies only publish pre-trained models, or users are unwilling to upload data when using the model compression service. Moreover, even given access to the training data, the computational costs of retraining models are often prohibitive in deployment scenarios. The inaccessibility of training data and the computational costs of retraining present obstacles to their widespread utilization.

In this work, we study the challenging task of pruning pre-trained networks without retraining, utilizing only a small unlabeled calibration set. Indeed, post-training compression has been widely

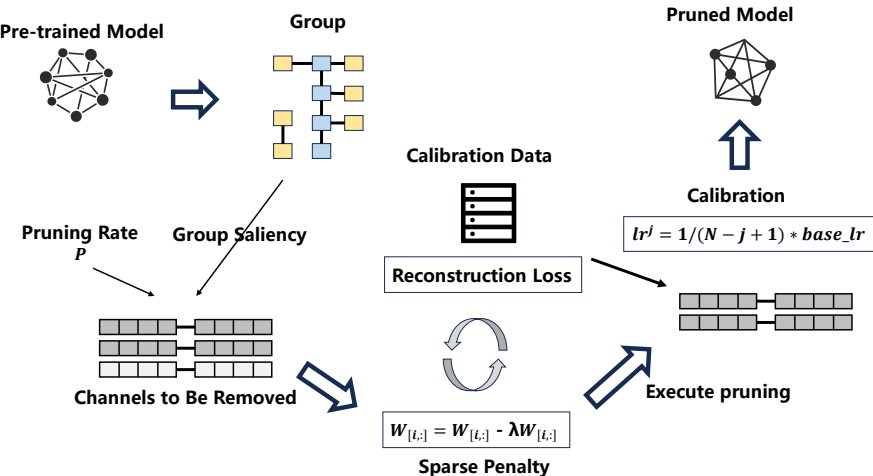

Figure 1: A visual overview of our post-training structured pruning framework. It outputs pruned models within a few minutes.

studied for quantization with promising results (Nagel et al., 2020; Hubara et al., 2021b; Li et al., 2021). These methods decompose the quantization task into layer-wise sub-problems, reconstructing each layer's output to approximate the task loss degeneration based on the calibration data. Due to its retraining-free advantages, post-training quantization is often preferred over quantization-aware training in practice. Recently, AdaPrune (Hubara et al., 2021a) demonstrated that this calibration approach can also be effective for unstructured pruning. However, effective post-training structured pruning is still an open problem.

The inherent nature of structured pruning makes post-training structured pruning of modern deep neural networks challenging. The complete removal of channels results in the inability to fully recover the intermediate representational features of the original model. As errors propagate across layers, we must handle compound errors. Moreover, complex inter-layer dependencies necessitate the simultaneous pruning of coupled channels to maintain network structure correctness. However, inconsistent sparsity distributions among coupled layers may result in removing channels encoding valuable information during pruning.

To address these challenges, we propose a novel post-training structured pruning framework that does not require retraining. As illustrated in Figure 1, our proposed framework takes a pre-trained model and a predefined pruning ratio as inputs. It firstly groups the network parameters according to layer dependencies and comprehensively gauges the grouped salience of coupled layers to inform pruning (see Section 3.1). Subsequently, a two-phase layer reconstruction procedure is conducted. The first phase intends to mitigate information loss before pruning. Since coupled layers are pruned simultaneously and expect consistent sparsity across layers, we impose an increasing regularization penalty on less important channels to condense representations into the retained components. The second phase executes pruning and calibrates the pruned model on a small calibration set to recover the accuracy (see Section 3.2). Our framework avoids expensive retraining, generally completing the whole process within minutes.

Our contributions can be summarized as follows:

- We propose a novel framework for efficient structured pruning in a post-training setting where only a small amount of unlabeled data is available. It integrates systematic pipeline components, including coupled channel pruning and a two-phase reconstruction procedure.

- Post-training structured pruning presents challenges due to the inevitable propagation of errors across layers and the inconsistent sparsity between coupled layers. To address these problems, the proposed two-phase reconstruction procedure first sparsifies channels scheduled for removal to force consistent sparsity before pruning. It then performs a holistic

reconstruction of the layer outputs to optimize composite error and recover the representation capacity of the original model.

- We conduct extensive experiments to validate the efficiency and effectiveness of our framework. Using merely about 0.2% of the ImageNet training set, our framework achieves 72.58% top-1 accuracy pruning ResNet-50, reducing FLOPs by 42%, comparable to full-data retraining methods. Notably, our framework generates pruned models within a few minutes, which is over 100x faster than the retraining-based methods.

## 2 RELATED WORK

**Network Pruning and Sparsity.** Pruning, as one of the popular network compression technique, has been widely studied (Hassibi et al., 1993; Han et al., 2015; He et al., 2017; Li et al., 2017). Mainstream pruning approaches can be broadly categorized into structured and unstructured methods. Unstructured pruning (Lee et al., 2020; Sanh et al., 2020; Frantar & Alistarh, 2022b) zeros out individual weights without altering the network structure, maintaining performance even at higher sparsity levels. However, unstructured pruning often necessitates specialized hardware or software for actual acceleration. In contrast, structured pruning (Liu et al., 2017; Molchanov et al., 2017; He et al., 2019) removes entire structured components from networks, thereby accelerating inference across diverse hardware.

**Pruning Coupled Structure.** Recent studies have developed methods for structured pruning of complex network architectures that contain coupled components that must be pruned simultaneously. While existing techniques utilize empirical rules or predefined architectural patterns to handle coupled structures Li et al. (2017); Liu et al. (2021); Luo & Wu (2020), these approaches have limited generalizability to new network topologies. Recent advances have focused on automated analysis of layer dependencies to tackle general structural pruning of arbitrary architecture. Narshana et al. (2023) introduced the notion of Data Flow Couplings (DFCs) to characterize couplings by enumerating the associated layers and transformations. Meanwhile, Fang et al. (2023) devised a dependency graph framework to capture layer associations and dependencies in a generalized manner. This work adopts a dependency graph approach (Fang et al., 2023) to identify coupled structures for structured pruning.

**Post-training Pruning.** Pruning without retraining is gaining interest owing to privacy concerns and computational costs associated with fine-tuning. Early efforts in this regime merged similar neurons to prune networks (Srinivas & Babu, 2015; Yvinec et al., 2021; Kim et al., 2020). These methods did not consider complex structural dependencies and failed to preserve performance under high compression ratios. Inspired by post-training quantization techniques, Hubara et al. (2021a); Frantar & Alistarh (2022b;a) solved a layer-wise reconstruction problem to minimize output change on calibration data. However, these approaches only support unstructured pruning and N: M sparse mode. Tang et al. (2020) directly developed new compact filters for structured pruning using original filters in their proposed Reborn technique. Although achieving acceptable accuracy under moderate sparity, the accuracy drop became larger with high sparsity. Kwon et al. (2022) proposed a post-training structured pruning method for Transformers, but it did not extend to convolutional networks due to the lack of consideration of complex multi-branched structures. Our work addresses the challenge of structured pruning of modern deep convolutional networks under limited data constraints.

## 3 METHOD

### 3.1 PRUNING COUPLED STRUCTURES

As insufficient data precludes retraining, correctly removing channels with minimal damage becomes imperative when pruning coupled channels. This highlights the precise measurement of channel importance in complex networks. We adopt the dependency defined in (Fang et al., 2023), abstracting both the layers and the non-parameterized operations in a network.

The network $F$ is formalized as $F = \{f_1, f_2, ..., f_l\}$, where $f_i$ refers to either a parameterized layer or a non-parameterized operation. The input and output of component $f_i$ are denoted as $f_i^-$ and $f_i^+$, which represent different pruning schemes for the same component. Dependencies are

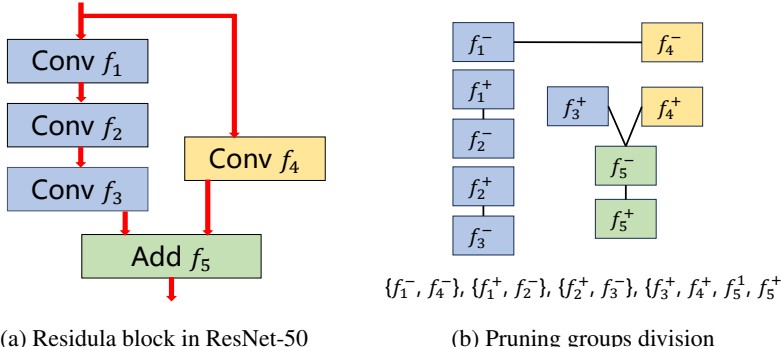

(a) Residula block in ResNet-50

(b) Pruning groups division

Figure 2: An instance of a residual block in ResNet-50 with a downsample layer. We show its pruning groups on the right.

categorized as inter-layer, determined by adjacent layer connectivity, or intra-layer, where input and output share pruning schemes, such as residual adding. By traversing all components, coupled structures are automatically analyzed and divided into different pruning groups, allowing tailored pruning schemes. Figure 2 shows an example of dividing pruning groups of a residual block in ResNet-50.

**Group Importance Estimation** Existing works have proposed algorithms for estimating the importance of individual channels, such as norm-based criterion (Li et al., 2017), channel similarity (He et al., 2019), and first-order Taylor error (Molchanov et al., 2019). However, as coupled layers within the same group should be pruned simultaneously, it is necessary to estimate the group-level importance. We first estimate the importance of individual channels using existing metrics. Due to the data limit, we only compare data-free measures: $L_1$, $L_2$, cosine distance, euclidean distance, and random, which we compare in the appendix. In this work, we select the $L_2$-norm-based criterion to estimate individual channel importance. Next, individual importance scores are aggregated to evaluate group-level importance. We also compare different aggregation ways and decide to aggregate scores by average.

## 3.2 Two-phase Layer Reconstruction Strategy with Holistic Layer Reconstruction Loss

Recently, several works (Frantar & Alistarh, 2022b; Hubara et al., 2021a) suggested lightweight calibration techniques, such as layer-wise reconstruction, to minimize the discrepancy between original and pruned model layer outputs:

$$\min_{\tilde{\mathbf{W}}} ||f^{(l)}(\mathbf{W}, \mathbf{X}) - f^{(l)}(\tilde{\mathbf{W}}, \tilde{\mathbf{X}})|| \tag{1}$$

where $f^{(l)}(\mathbf{W}, \mathbf{X})$ and $f^{(l)}(\tilde{\mathbf{W}}, \tilde{\mathbf{X}})$ are the output of layer $l$ of the original model and pruned model respectively. These methods achieve impressive performance in unstructured pruning settings. However, their direct application to structured pruning settings does not produce satisfying results (see Table 2). We argue that this stems from their failure to consider the inherent properties of the structured pruning paradigm. To adjust calibration techniques into structured settings, we extensively investigate the inherent properties of structured pruning and summarize as follows:

- **Grouped layers exhibit structural dependency:** In the structured pruning setting, layers are not independently pruned. Previous work by Hubara et al. (2021a) optimized the output of each layer independently, thereby shielding the influence of the accumulated errors propagated from previous layers.

- **Layers are grouped while information is scattered:** The sparsity of different layers is inconsistent (Narshana et al., 2023; Fang et al., 2023) in a pruning group. Some removed channels may still encode useful information. We calculate the grouped $L_2$-norm distribution of a pre-trained ResNet-50. As shown in Figure 3(a), even if 30% of the channels are removed, we prune many channels that contain useful information. Since output re-

construction can only obtain information from the remaining channels/filters in the original model, the information in the pruned channel is difficult to recover.

The first property indicates that the objective function of layer reconstruction should be holistic. The second property suggests that information should be concentrated and coherent with the grouped structure before pruning to enable superior pruning performance. To this end, we propose a holistic layer reconstruction loss with the layer-variant update rate to improve the independent layer-wise modeling of (1). We then propose a two-phase layer reconstruction strategy where we concentrate the information via $L_2$-penalized optimization as a pre-reconstruction step. The details of our approach are elaborated in the subsequent sections.

### 3.2.1 HOLISTIC LAYER RECONSTRUCTION LOSS

In structured pruning, the number of input and output channels is reduced for most layers, thereby restricting the features available for reconstructing the original layer outputs. The cumulative impact of previous layers cannot be ignored, especially when pruning channels in complex networks. To model this impact, we employ a compounding reconstruction loss to jointly optimize each layer in context based on the remaining channels:

$$\mathcal{L}_{re}(\tilde{\mathbf{W}}) = \sum_{l=1}^{L} ||[f^{(l)}(\mathbf{W}, \mathbf{X})]_{\mathbf{s}^{(l)},:,:,:} - f^{(l)}(\tilde{\mathbf{W}}, \tilde{\mathbf{X}})||_2^2 \tag{2}$$

where $\mathbf{s}^{(l)}$ indexes the remaining output channels of layer $l$. Eq. 2 extracts output features corresponding to the retained channels in the original model to calculate the reconstruction loss. Naive optimizing Eq. 2 induces imbalanced parameter updates, with earlier layers being over-optimized. We normalize the gradient norm of each layer by their weight norm to fairly compare updates at each layer and visualize layer-wise update rates as follows:

$$R = lr * \frac{||\mathbf{G}||_2}{||\mathbf{W}||_2} \tag{3}$$

where $\mathbf{G}$ and $\mathbf{W}$ are the gradient and weight of a layer. To mitigate this imbalance, we propose scaling the learning rate layer-wise as $1/(L - j + 1) * base\_lr$ for layer $j$, where $L$ is the total number of layers. Our ablation studies demonstrate that this calibrated rate schedule significantly improves output reconstruction, substantiating our approach. (See Section 4.3).

### 3.2.2 TWO-PHASE LAYER RECONSTRUCTION PROCEDURE

In order to produce consistent sparsity across grouped layers for better pruning, we propose a two-phase layer reconstruction strategy. The key idea is to consolidate the information in the channels scheduled for removal through a pre-reconstruction step before actually pruning them. Given a pre-trained model with parameters $w$ and the reconstruction error, we formulate this problem as a structural $L_2$-penalized regression problem as follows:

$$\mathcal{L}_{sp}(\tilde{\mathbf{W}}) = \mathcal{L}_{re}(\tilde{\mathbf{W}}) + \frac{1}{2} \sum_{l,i} \lambda_i^{(l)} ||\tilde{\mathbf{W}}_{i,:,:}^{(l)}||_2^2 \tag{4}$$

where $\mathcal{L}_{re}$ is the holistic reconstruction loss (Eq. 2) as fidelity term, $\tilde{\mathbf{W}}_{i,:,:}^{(l)}$ is the $i$-th channel of the weight of layer $l$, and $\lambda_i^{(l)}$ is the $L_2$ regularization co-efficient for that channel. $\lambda_i^{(l)} > 0$ if the channel will be removed. Otherwise, $\lambda_i^{(l)} = 0$ since we do not penalize reserved weights. We gradually increase the penalty strength to mitigate the side effect of the sparse penalty: $\lambda = \lambda + \delta$, where $\delta$ is a pre-defined constant. By gradually increasing the penalty intensity, we can concentrate the information into the retained channels while sparsifying the unimportant ones.

After this specification phase, the unimportant channels exhibit high sparsity and can be safely removed (See Figure 3(b)). We track the $L_2$-norm trends of the reserved and soon-to-be pruned channels over each iteration in Figure 3(c). The norm sum of remaining channels maintains a high level, and those to be pruned drop steadily to very small values, which indicates the informative features are squeezed into the remaining channels. Because the information is compactly concentrated in the first phase of the procedure, the second phase is much more trivial and just optimizes holistic layer reconstruction loss to achieve good results.

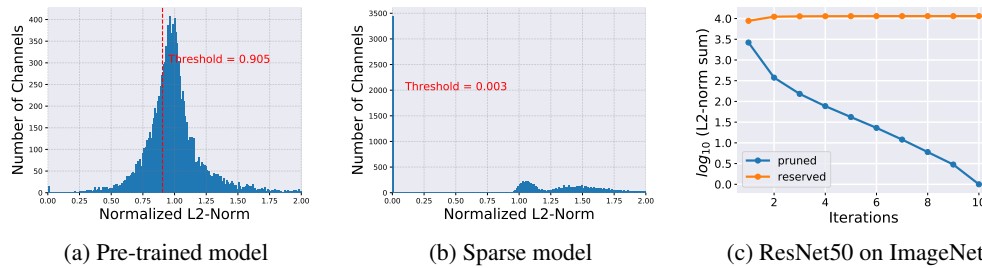

(a) Pre-trained model      (b) Sparse model      (c) ResNet50 on ImageNet

Figure 3: (a & b) Distribution of $L_2$-norm of weights for each grouped channel in a ResNet-50 model. The channel norm of each pruning group is normalized by the mean of the norms. The statistical data in (a) and (b) come from the pre-trained model and the model after the sparsity penalty, respectively. We marked the threshold at a pruning rate of 0.3. (c) The $L_2$-norm sum of reversed channels and those to be pruned (logarithmic scale).

## 3.3 POST-TRAINING STRUCTURED PRUNING

Algorithm 1 summarizes the pruning process of our framework. The layers in the copy of a pre-trained model are first partitioned into pruning groups based on layer dependencies. Then the group-level importance of channels is estimated across layers within each group. Channels with the lowest importance scores, up to a predefined pruning ratio $r$, are marked for removal. Before removing these marked channels, channel sparsification is performed by minimizing Eq. 4 over several iterations. This transfers the expressive capacity of the less important channels to the remaining components. Upon completion of the sparsification process, the marked channels are pruned to obtain a compact model. Finally, the original model output is reconstructed to recover accuracy.

---

**Algorithm 1** PSP: Post-training Structured Pruning

---

**Input:** A pre-trained model $\mathcal{F}_1(\mathbf{W})$, pruning ratio $p$, calibration dataset, penalty coefficient $\lambda$ and its increment $\delta$, iteration $T_s,T_r$.
**Init:** Copy $\mathcal{F}_1(\mathbf{W})$ to form $\mathcal{F}_2(\tilde{\mathbf{W}})$.
**Init:** Divide the layers in model $\mathcal{F}_2(\tilde{\mathbf{W}})$ into different pruning groups.
**Init:** Mark indexes of channels to be pruned by group-level $L_2$-norm sorting.
**for** $i = 0$ **to** $T_s - 1$ **do**
    Compute reconstruction loss $\mathcal{L}_{re}(\tilde{\mathbf{W}})$ on calibration dataset and update model $\mathcal{F}_2(\tilde{\mathbf{W}})$:
    $\tilde{\mathbf{W}} \leftarrow \tilde{\mathbf{W}} - \eta(\nabla_{\tilde{\mathbf{W}}}\mathcal{L}_{re}(\tilde{\mathbf{W}}) + \lambda\tilde{\mathbf{W}})$
    $\lambda \leftarrow \lambda + \delta$
**end for**
Prune marked channels of the model $\mathcal{F}_2(\tilde{\mathbf{W}})$.
**for** $j = 0$ **to** $T_r - 1$ **do**
    Compute reconstruction loss $\mathcal{L}_{re}(\tilde{\mathbf{W}})$ on calibration dataset and update model $\mathcal{F}_2(\tilde{\mathbf{W}})$.
    $\tilde{\mathbf{W}} \leftarrow \tilde{\mathbf{W}} - \eta(\nabla_{\tilde{\mathbf{W}}}\mathcal{L}_{re}(\tilde{\mathbf{W}}))$
**end for**
**Output:** Pruned model $\mathcal{F}_2(\tilde{\mathbf{W}})$.

---

## 4 EXPERIMENTS

### 4.1 SETTINGS

**Datasets and architectures.** We conducted extensive experiments on three representative datasets, CIFAR-10, CIFAR-100 (Krizhevsky et al., 2009), and ImageNet (Deng et al., 2009). For CIFAR datasets, we train the base models from scratch with a batch size of 128, and the learning rate schedule initialized as 0.1, multiplied by 0.1 at epoch 50 and 75 for total 100 epochs. For ImageNet, we use the official PyTorch (Paszke et al., 2019) pre-trained models as the base models.

Table 1: Pruning results on CIFAR-10/100. RP denotes the reduction in parameters, and RF denotes the reduction in FLOPs. Due to the random sampling during calibration, we report the average accuracy over 10 runs.

| Method | CIFAR-10 | | | CIFAR-100 | | |
|---|---|---|---|---|---|---|
| | Acc(%) | RP | RF | Acc(%) | RP | RF |
| ResNet-50 | 94.36 | - | - | 78.86 | - | - |
| $L_1$-norm (retrained) | 92.80 | 2.36x | 1.94x | 74.58 | 2.22x | 3.02x |
| Neuron Merging | 87.24 | 2.19x | 1.76x | 65.27 | 1.64x | 2.03x |
| AdaPrune | 93.04 | 4.00x | 3.42x | 67.52 | 3.32x | 4.33x |
| gAP | 93.58 | 4.00x | 3.42x | 68.34 | 3.32x | 4.33x |
| Reborn Filters | 91.56 | 2.58x | 2.15x | 70.40 | 1.75x | 1.90 x |
| PSP (Ours) | **94.01** | **4.00x** | **3.42x** | **75.16** | **3.32x** | **4.33x** |
| VGG-19 | 93.01 | - | - | 72.27 | - | - |
| $L_1$-norm (retrained) | 91.57 | 3.56x | 2.14x | 69.65 | 2.57x | 1.67x |
| Neuron Merging | 90.70 | 2.85x | 1.50x | 67.63 | 2.86x | 1.74x |
| AdaPrune | 90.55 | 4.30x | 2.67x | 68.53 | 3.02x | 1.82x |
| gAP | 91.69 | 4.30x | 2.67x | 68.37 | 3.02x | 1.82x |
| Reborn Filters | 90.46 | 3.87x | 2.25x | 69.03 | 2.33x | 1.54x |
| PSP (Ours) | **92.15** | **4.30x** | **2.67x** | **70.58** | **3.02x** | **1.82x** |

**Implementation details.** We randomly sample 512 images from the training set for CIFAR-10/100 and 2048 images for ImageNet for pruning,. We do not use any data augmentation. The batch size is set to 128 for CIFAR and 64 for ImageNet. The sparsity regularization coefficient $\lambda$ in Eq.4 is initialized to 0.02 and increased by 0.02 each iteration. We optimize all models using the ADAM optimizer with a base learning rate $10^{-3}$. The number of iterations for sparsity penalty and output reconstruction is 20 and 10, respectively.

**Baselines.** We baseline our work against diverse representative retraining-free pruning approaches to evaluate the effectiveness of our pruning framework. Specifically, we make comparisons to the following retraining-free techniques: (i) Neuron Merging (Kim et al., 2020), which compensates for the information loss from pruned neurons/filters by merging similar neurons. (ii) AdaPrune (Hubara et al., 2021a), which independently reduces the discrepancy between the pruned and original layer outputs. (iii) gAP (Frantar & Alistarh, 2022b), a of variants AdaPrune. (iv)Reborn Filters (Tang et al., 2020), which directly constructs new compact filters using the original filters.AdaPrune and gAP are designed for unstructured pruning, and we extend them to structured pruning. We implement AdaPrune and gAP using the same group importance estimation as our method. Other baselines use the channel importance evaluation method presented in their paper. We also compare with retraining-based methods to show the gap with them, including $L_1$-norm-based pruning (Li et al., 2017), DepGraph (Fang et al., 2023) and GReg-2 (Wang et al., 2021).

## 4.2 PSP CAN EFFECTIVELY RESTORE PERFORMANCE

Our proposed approach demonstrates consistent and notable improvements over other retraining-free pruning techniques on the CIFAR-10 and CIFAR-100 datasets, as shown in Table 1. Specifically, utilizing a ResNet-50 architecture pruned to over 3x reduction in FLOPs, our method achieves an accuracy of 94.01% on CIFAR-10 and 75.16% on the more complex CIFAR-100, outperforming existing retraining-free techniques with similar parameter and FLOP budgets. Compared to AdaPrune and gAP, the advantages of our framework are particularly pronounced on the intricate CIFAR-100 task, highlighting the importance of concentrating information before pruning. Surprisingly, our framework attains equivalent or superior accuracy relative to the retraining pruning baseline, empirically evidencing that retraining may be obviated in pruning pipelines, at least for small datasets. Besides, techniques using no data, like Neuron Merging, suffer from huge accuracy drops and cannot be applied to higher sparsity. This indicates that a small amount of calibration data is necessary to maintain accuracy. Overall, our proposed post-training structured pruning framework demonstrates highly competitive performance.

Table 2: Pruning ResNet-50 trained on ImageNet. We randomly sampled 2048 samples from the training set as calibration data. RP denotes the reduction in parameters, and RF denotes the reduction in FLOPs

| Method | Retraining? | Acc(%) | RP | RF | Acc(%) | RP | RF |
|---|---|---|---|---|---|---|---|
| ResNet-50 | - | 76.13 | - | - | 76.13 | - | - |
| $L_1$-norm | Yes | 72.27 | 1.45x | 1.82x | 71.32 | 1.78x | 2.55x |
| DepGraph | Yes | 75.83 | - | 2.07x | - | - | - |
| GReg-2 | Yes | 75.36 | - | 1.50x | - | - | - |
| Neuron Merging | No | 39.58 | 1.36 | 1.67x | - | - | - |
| AdaPrune | No | 62.49 | 1.38x | 1.73x | 50.97 | 1.66x | 2.36x |
| gAP | No | 56.88 | 1.38x | 1.73x | 54.30 | 1.66x | 2.36x |
| Reborn Filters | No | 59.08 | 1.22x | 1.49x | 57.11 | 1.54x | 2.13x |
| PSP (Ours) | No | **72.58** | **1.38x** | **1.73x** | **69.85** | **1.66x** | **2.36x** |

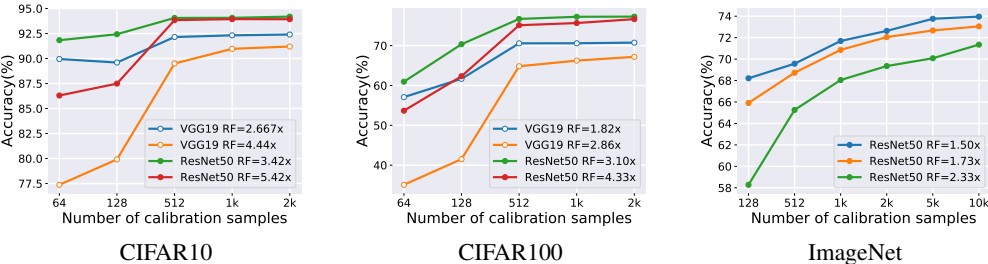

|  | CIFAR10 | CIFAR100 | ImageNet |

Figure 4: Accuracy of pruning pre-trained models on CIFAR-10, CIFAR-100, and ImageNet w.r.t. different number of calibration samples. We plot results with different FLOPs reductions.

Table 2 presents the pruning results of ResNet-50 on the ImageNet dataset. This work aims to prune pre-trained models when only a small amount of data is available. As such, we only use $L_1$-norm pruning as a baseline to demonstrate the gap compared to expensive retraining methods. Due to the scale of ImageNet, the model has little channel redundancy, and structured pruning unavoidably discards valuable information. While techniques such as AdaPrune and gAP have shown acceptable accuracy degradation on smaller datasets like CIFAR, their utility decreases for large-scale datasets like ImageNet. In contrast, our proposed framework can recover accuracy using only 0.2% of the ImageNet training data, demonstrating efficacy even for large-scale datasets. In particular, our framework attains 72.58% top-1 accuracy and a 1.73x reduction in FLOPs using only 2048 samples and a few iterations. By using sparse penalty and holistic reconstruction, we can achieve comparable performance without retraining, even on large-scale datasets.

### 4.3 ABLATION STUDY

**Impact of the number of calibration samples.** We conduct experiments incrementally increasing the number of calibration samples to further investigate the influence of varying calibration sample sizes on our pruning framework. The results are presented in Figure 4. Overall, model accuracy tends to improve as more calibration data is used. This trend is expected, as additional calibration samples provide the pruned model with greater information about the original model's behavior. The rate of accuracy improvement decreases and begins to stabilize as the calibration set size approaches 1% of the training data for the CIFAR datasets. For the ImageNet dataset, the accuracy can still be improved when the amount of data increases from 5k to 10k. We finally achieved 73.96% top-1 accuracy with a 1.50x FLOPs reduction using 10k calibration data. Notably, our approach maintains acceptable accuracy even when very limited calibration data is available. For instance, with only 64 samples from CIFAR-10, ResNet-50 pruned by our method retains 91.9% accuracy while reducing FLOPs to 29%.

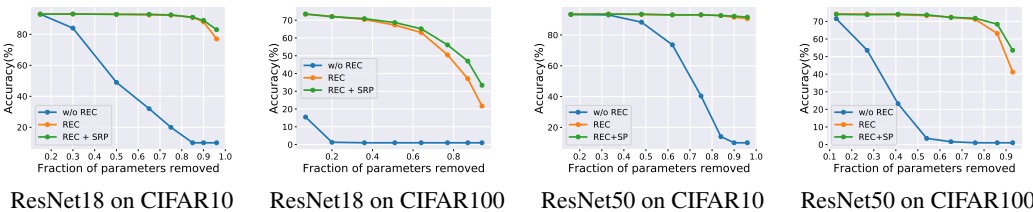

Figure 5: Figures comparing Accuracy versus Sparsity (through parameters) for our method. We use the corresponding parameter removal ratio as the horizontal axis. In each figure, the plots show the performance under three conditions: doing nothing (w/o REC), only reconstruction (REC),and reconstruction with sparse penalty (REC+SP).

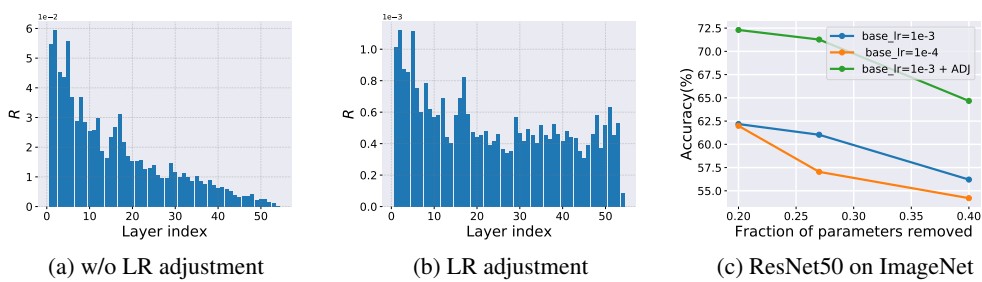

Figure 6: (a)Update rates for convolutions and fully-connected layers in a ResNet-50 model. Statistics are accumulated over 1024 samples. (b) Updates become balanced through the layer-wise learning rate adjustment. (c) Accuracy versus Sparsity with ResNet-50 on ImageNet. In order to exclude the influence of other factors, we did not perform sparsity penalty.

**Effect of sparse penalty and holistic reconstruction.** In order to investigate the effects of the sparse penalty and holistic reconstruction, we conduct an ablation study across a range of pruning rates from 0.1 to 0.8 in increments of 0.1. Given the potential for irrecoverable damage from aggressive one-shot pruning at high sparsity levels, we employed an iterative approach with five iterations. Figure 5 demonstrates that at low pruning rates, holistic reconstruction alone is sufficient to recover model accuracy, owing to the presence of redundant channels. However, as the pruning rate increases, the benefit of sparse penalty becomes evident, indicating it compressed the information from removed channels into the remaining components.

**Update rate balance.** As shown in Figure 6(a), we observe that update rates show a decreasing trend as the layer depth increases. However, by using our proposed layer-wise learning rate calibration strategy, update rates across layers become more balanced, as shown in Figure 6(b). To exclude the effect of smaller learning rates, we also evaluate the accuracy when the base learning rate is reduced by a factor of 10 in Figure 6(b). The results demonstrate that our approach significantly improves the performance of output reconstruction.

## 5 CONCLUSION

This work proposes an effective framework for structured pruning of networks with limited data. Our framework first groups coupled channels by layer dependencies and determines pruning priorities based on group-level importance. Sparse regularization is then applied to the channels scheduled for removal, concentrating information into the remaining channels before pruning. Our framework finally reconstructs the original layer outputs using the preserved channels of the pruned network. Experiments demonstrate remarkable improvements against baseline methods in a post-training setting. Notably, when using only 1% of the ImageNet training data, our framework reduced FLOPs in ResNet-50 by 1.73x with just a 3.55% drop in accuracy. By enabling highly efficient structured pruning without expensive retraining, our work provides an important advancement for pruning models in deployment scenarios.

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

# A MORE EXPERIMENTS AND ABLATION STUDIES

## A.1 TIME COST OF OUR FRAMEWORK

**Hardward and software details.** The time cost measurements are performed using a Tesla V100 with CUDA 11.7 and a memory of 16GB. The software stack used for inferencing consisted of Python 3.9.16, PyTorch 1.13.1, and Torchvision 0.14.1.

We systematically analyze the time cost of our framework to produce a pruned model. As shown in Table 3, our framework requires only about 43 seconds to prune ResNet-50 on CIRAR-10 and 8 minutes to prune ResNet-50 on ImageNet. In contrast, retraining-based approaches typically take dozens of hours to complete pruning on ImageNet, which is 2-3 orders of magnitude slower. We highlight that the little time consumption and data requirements make it feasible to apply pruning at deployment time.

Table 3: Pruning costs of our framework. We report the time cost (in seconds) of different phases as well as the whole pipeline. The results are tested on a single Tesla V100.

| Model | Dataset (number of samples) | Sparse penalty | Reconstruction | All |
|---|---|---|---|---|
| ResNet-18 | | 14.4 | 6.0 | 21.6 |
| ResNet-50 | CIFAR-10 (512) | 30.5 | 11.1 | 42.8 |
| VGG-19 | | 12.8 | 5.2 | 19.3 |
| ResNet-50 | ImageNet (2k) | 337.9 | 127.4 | 467.4 |

## A.2 COMPARISONS OF DIFFERENT CRITERIA AND AGGREGATION APPROACHES

To investigate the effectiveness of different criteria for ranking and pruning channels at a group level, we evaluate five criteria for pruning ResNet-18 on the CIFAR-10 dataset, including random, $L_2/L_1$-norm of weights, cosine distance, and Euclidean distance. In order to exclude the influence of other operations, we fix the aggregation method to the averaging and compare the pruning accuracy across criteria without output reconstruction. As shown in Figure 7(a), $L_2$-norm demonstrates strong performance at all sparsity levels and slightly outperforms other criteria at high sparsities. This suggests that $L_2$-norm is an effective metric for producing independent scores of channel importance in group-level pruning.

We further compare three different aggregation strategies, including (i) Mean: $I_g = \frac{1}{N} \sum_{i=1}^{N} I_{W_i}$, (ii)Max: $I_g = \max_{i=1}^{N} I_{W_i}$, and (iii)Product: $I_g = \prod_{i=1}^{N} I_{W_i}$, where $I_W$ is L2-norm magnitude of each $W \in g$. As shown in Figure 7(b), mean and max aggregation yield similar accuracy, while product aggregation does not produce meaningful accuracy. This implies the maximum value in

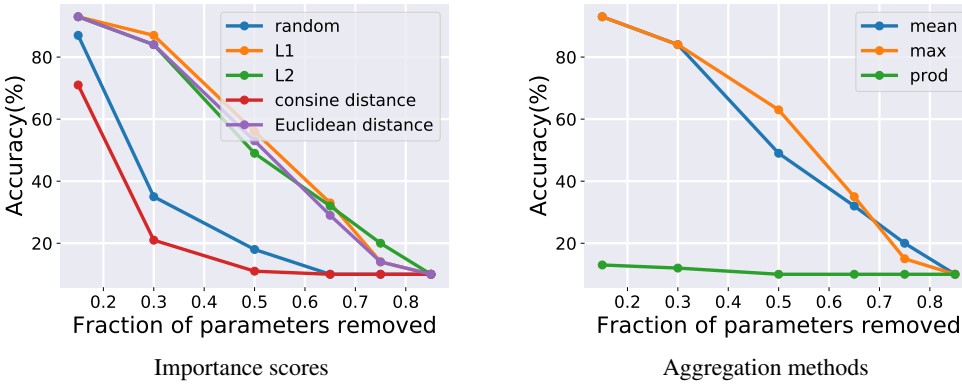

Figure 7: Figures comparing Accuracy versus Sparsity (through parameters) for different criteria and aggregation methods. We prune ResNet-18 trained on CIFAR-10 under pruning rates selected from {0.1, 0.2, 0.3, 0.4, 0.5, 0.6} and use the corresponding parameter removal ratio as the horizontal axis.

Table 4: Pruning results of various models on ImageNet. We randomly sampled 2048 samples from the training set as calibration data. The pruning rates are set to 0.2 and 0.3, respectively.

| Method | Acc-1(%) | RP | RF | Acc-1(%) | RP | RF |
|---|---|---|---|---|---|---|
| **ResNet101**: Baseline accuracy 77.37%, #Params: 44.55M, FLOPs: 7.85G | | | | | | |
| AdaPrune | 70.49 | | | 66.69 | | |
| gAP | 60.09 | 1.38x | 1.59x | 57.49 | 1.76x | 2.14x |
| PSP (Ours) | **73.84** | | | **72.52** | | |
| **GoogLeNet**: Baseline accuracy 69.78%, #Params: 6.62M, FLOPs: 1.51G | | | | | | |
| AdaPrune | 52.23 | | | 34.83 | | |
| gAP | 54.37 | 1.40x | 1.84x | 40.15 | 1.76x | 2.36x |
| PSP (Ours) | **60.27** | | | **51.38** | | |
| **MobileNetV2**: Baseline accuracy 71.88%, #Params: 3.5M, FLOPs: 0.32G | | | | | | |
| AdaPrune | 45.00 | | | 10.00 | | |
| gAP | 61.25 | 1.14x | 1.45x | 51.47 | 1.24x | 1.78x |
| PSP (Ours) | **66.74** | | | **60.06** | | |
| **EfficientNetV2**: Baseline accuracy 81.31%, #Params: 21.46M, FLOPs: 2.88G | | | | | | |
| AdaPrune | 70.97 | | | 51.59 | | |
| gAP | 68.79 | 1.26x | 1.16x | 64.33 | 1.45x | 1.27x |
| PSP (Ours) | **73.35** | | | **69.51** | | |

each channel of a pruning group may determine whether that channel is pruned. In contrast, product aggregation is affected by small values in coupled channels, failing to correctly estimate group-level channel importance. This result instructs us to prune channels whose all parameters are consistently unimportant. The sparsification process of our framework produces strong sparsity at the group level, which is beneficial for pruning coupled channels.

## A.3 PRUNE VARIOUS MODELS ON IMAGENET

We evaluate the scalability of our framework to different network architectures in Table 4. The results demonstrate that the proposed framework achieves superior performance compared to AdaPrune and gAP baselines across diverse convolutional neural network architectures. An interesting observation is that pruning tended to remove channels with small parameters and computational complexity for EfficientNetV2 while removing larger channels for GoogLeNet. This may

Table 5: Comparisons of one-shot pruning and iterative pruning with ResNet-50 trained on CIFAR-10 and CIFAR-100. We randomly sampled 512 samples from the training set as calibration data.

| **ResNet50 on CIFAR10**: Baseline accuracy 94.36%, #Params: 23.52M, FLOPs: 1.30G | | | | | |
| --- | --- | --- | --- | --- | --- |
| Pruning ratio | 0.4 | 0.5 | 0.6 | 0.7 | 0.8 |
| #Params (M)/FLOPs (G) | 8.40/0.55 | 5.88/0.38 | 3.79/0.24 | 2.12/0.13 | 0.95/0.05 |
| Acc. (% One-shot) | 94.10 | 94.01 | 93.69 | 92.17 | 90.14 |
| Acc.(% Iteration-5) | 94.13 | 94.08 | 93.84 | 92.62 | 91.05 |
| Acc.gain(%) | 0.03 | 0.07 | 0.15 | 0.45 | 0.91 |
| **ResNet50 on CIFAR100**: Baseline accuracy 78.86%, #Params: 23.71M, FLOPs: 1.30G | | | | | |
| Pruning ratio | 0.4 | 0.5 | 0.6 | 0.7 | 0.8 |
| #Params (M)/FLOPs (G) | 10.68/0.42 | 7.13/0.30 | 4.14/0.21 | 1.99/0.13 | 0.77/0.07 |
| Acc.(% One-shot) | 77.07 | 75.16 | 70.74 | 56.92 | 24.38 |
| Acc.(% Iteration-5) | 77.18 | 75.20 | 71.39 | 68.40 | 53.60 |
| Acc.gain(%) | 0.11 | 0.04 | 0.65 | 11.48 | 29.22 |

be attributed to the multi-branch nature of GoogLeNet, causing simultaneous pruning across layers. We also observe that even when pruning lightweight networks like MobileNetV2, our framework achieves acceptable accuracy degradation. Overall, our framework achieves consistent accuracy improvements over baselines across various networks, reflecting our framework's generalization ability.

## A.4 ITERATIVE POST-TRAINING STRUCTURED PRUNING

Algorithm 2 summarizes the iterative pruning version for our post-training structured framework. The algorithm progressively removes less important channels over multiple iterations. In each iteration, the framework recalculates channel importance and performs a two-stage reconstruction process. This iterative structured pruning procedure circumvents a sharp drop in model accuracy that can occur when an excessive number of channels are removed simultaneously. Table 5 shows the comparisons of one-shot and iterative pruning applied to ResNet-50 models trained on the CIFAR10 and CIFAR100 datasets. On the relatively simple CIFAR-10 dataset, both techniques yield comparable performance across all sparsity levels, reflecting the high degree of redundancy within the network. However, as evidenced by the CIFAR100 results, iterative pruning achieves significant gains in preserving accuracy over one-shot pruning as the sparsity increases, with improvements of 11.48% and 29.22% attained at pruning rates of 70% and 80%, respectively. By gradually removing less important channels, the iterative approach largely overcomes catastrophic loss of information.

---

**Algorithm 2** Iterative Post-training Structured Pruning

---

**Input:** A pre-trained model $\mathcal{F}_1(\mathbf{W})$, pruning ratio $p$, calibration dataset, penalty coefficient $\lambda$ and its increment $\delta$, iteration $T_s, T_r, T$.
**Init:** Copy $\mathcal{F}_1(\mathbf{W})$ to form $\mathcal{F}_2(\tilde{\mathbf{W}})$.
**Init:** Divide the layers in the model $\mathcal{F}_2(\tilde{\mathbf{W}})$ into different pruning groups.
**for** $t = 0$ **to** $T$ **do**
    Mark indexes of channels to be pruned by group-level $L_2$-norm sorting.
    Doing sparse penalty over $T_s$ iterations
    Prune marked channels
    Doing output reconstruction over $T_r$ iterations
**end for**
**Output:** Pruned model $\mathcal{F}_2(\tilde{\mathbf{W}})$.

---

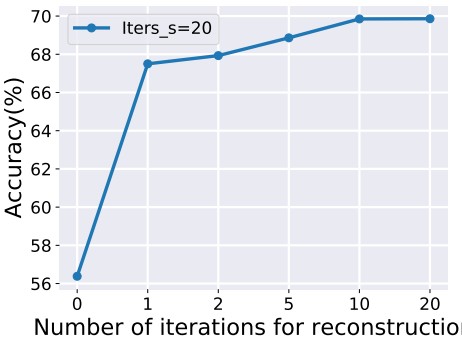 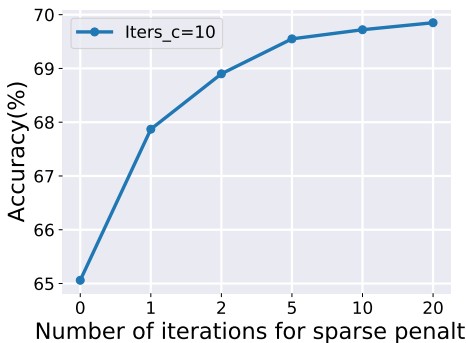

Figure 8: Accuracy of pruning pre-trained ResNet-50 on ImageNet dataset. We prune the model to a 1.73x FLOPs reduction. (Left)Accuracy of reconstructions performed with different iterations when the number of iterations of the sparsity penalty is fixed at 20. (Right) Accuracy of performing sparse penalty for different iterations when the number of iterations of reconstruction is fixed to 10.

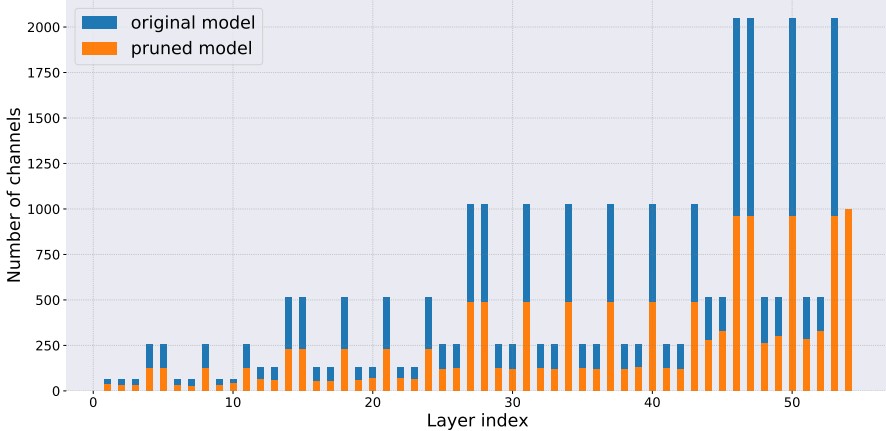

Figure 9: Width of layers in pruned models and original models. We prune 50% channels of a ResNet-50 trained on ImageNet.

### A.5 THE IMPACT OF THE NUMBER OF ITERATIONS FOR THE SPARSITY PENALTY AND RECONSTRUCTION

We further investigated the impact of the number of iterations for the sparsity penalty and reconstruction processes on model accuracy when pruning ResNet-50 on ImageNet. The results in Figure 8 demonstrate that increasing iterations for both processes improves accuracy, although benefits decrease after 10 reconstruction iterations. Specifically, even just one reconstruction iteration after 20 sparsity penalty iterations boosts accuracy by 11 percentage points compared to no reconstruction. This highlights the importance of the interplay between the two iterative processes for retaining accuracy post-pruning.

### A.6 THE PRUNED RESNET-50 ARCHITECTURE

In Figure 9, we present the remaining channels of a pruned ResNet-50. Except for the final fully connected layer, all layers have been pruned. We can also observe that coupled layers possess an equivalent number of remaining channels. This property ensures the pruned model can run correctly.

