# OpenReview forum: "A Fast Framework for Post-training Structured Pruning Without Retraining"
_ICLR.cc/2024/Conference — Submitted to ICLR 2024_

### Official Review · Reviewer_4NdE · 2023-10-30

**Soundness:** 2 fair
**Presentation:** 3 good
**Contribution:** 3 good
**Rating:** 5
**Confidence:** 4

**Summary:**

The paper proposes a method to alleviate accuracy drops observed upon (structured) pruning a (complex multi-branched) network without finetuning / retraining. The main idea is a sample efficient way to reduce perturbations of intermediate layers before pruning via jointly minimizing the reconstruction errors of intermediate outputs and increasing regularization iteratively for channels that are to be pruned.  This ensures that representational capability of the channels to be pruned are not significantly hampered by transferring them to other unpruned channels during the reconstruction process.

While not being completely sample-free, the method demonstrates superior performance to other weight reconstruction / calibration methods. Thus this work makes progress towards suggesting that complete finetuning may not be required at all in structured pruning pipelines if reconstruction based methodologies are able to recover the lost accuracies.

The idea of transferring representational capacity via increasing regularization is not entirely novel [a]. Nor is the idea of reducing the reconstruction errors of intermediate outputs [b, c, d]. However, their combination is quite interesting here.

[a] Neural pruning via growing regularizaiton. Wang et. al. 2021 ICLR.

[b] ThiNet: A Filter Level Pruning Method for Deep Neural Network Compression. Luo et al 2017.

[c] DFPC: DFPC: Data flow driven pruning of coupled channels without data. Narshana et al. 2023 ICLR

[d] Spdy: Accurate pruning with speedup guarantees. Frantar et al. 2022 ICML

**Strengths:**

S1. The problem of recovering accuracy without finetuning is important and challenging.

S2. The method seems technically sound and straightforward in principle

S3. Empirical results demonstrate the strength of this approach.

**Weaknesses:**

W1. In section 3.2, the authors claim that the direct application of layer-wise reconstruction to minimize the discrepancy between original and pruned model to structured pruning does not produces satisfying results. However, there are no experiments or citations to justify this claim - weakening the justification of the proposed methodology.

W2. The authors do not report the FLOPs and wall-clock time required by the post training structured pruning framework. Also, a comparision of compute with finetuning methods is missing since this is a motivation provided behind the work.

W3. The retraining based pruning baselines selected are weak. Also, information is not provided as to how are those pruned models obtained. For example, it is not clear how the L1-norm pruned model has been generated in Table 2. Moreover, [a], [b], [c] show much superior performance with respect to the L1-norm pruned baseline chosen upon retraining.

[a] Only Train Once: A One-Shot Neural Network Training And Pruning Framework. Chen et al. NeurIPS 2021.
[b] DFPC: Data flow driven pruning of coupled channels without data. Narshana et al. 2023 ICLR
[c] Neural Pruning via Growing Regularization. Wang et al. NeurIPS 2021

**Questions:**

Q1. How were the callibration samples selected for plots in Fugure 4? How sensitive are these plots do different set of callibration samples? Would the accuracy of the models change if the callibration samples change?

Q2. In Algorithm 1, how are the layer-wise pruning ratios selected? (Please note the typo in the last weight update of Algorithm 1 (third last line). Do you intend to minimize the output layer's reconstruction loss here?)

Q3. In section 3.2, the authors state that "the sparsity of different layers is inconsistent in a pruning group". Can you please elaboarate. It is unclear what you intend to say here.

---

> ### Author Response · Authors · 2023-11-14
> **Response to Reviewer 4NdE**
>
> Dear Reviewer 4NdE,
>
> We thank you for your valuable feedback and that you find strength in our problem statement and methodology. Sorry for any misunderstanding caused by our presentation and layout. We respond to your comments below.
>
> 1. **There are no experiments or citations to justify this claim (W1).**
>     * We extended Adaprune and gAP to directly apply layer-wise reconstruction on structured pruning, and experiments on ImageNet have shown that these methods cannot effectively alleviate the accuracy decrease caused by pruning (See Table 2). We will explicitly reference the results in the modified version.
>
> 2. **The authors do not report the FLOPs and wall-clock time required (W2).**
>     * We previously thought that the report on actual running time was not very important, so it was included in appendix A.1. We will consider presenting this result in the main text in the revised version. In addition, we have reported the reduction in FLOPs, so no specific FLOPs were displayed.
>
> 3. **The retraining-based pruning baselines selected are weak (W3).**
>     * Our paper focuses on pruning without retraining, so we only compare the $L_1$-norm based method to show the gap with retraining-based methods. We reproduced the pruning algorithm in [1] as $L_1$-norm pruned baseline and we retrained 100 epochs after pruning.
>     * We have added the results of [2] and [3] in the revision. By the way, DFPC ([4]) is proposed as a data-free method retrained, it does not generalize to higher sparsity. Moreover, it retrained the model when reporting results on ImageNet. This method does not match our settings and therefore was not selected for the baseline.
>
> 4. **More experimental details and sample sensitivity experiments (Q1\& Q2).**
>     * We randomly sample indexes to obtain random samples. And the selection of samples only causes minor changes in the results. The results of ten calibrations after pruning ResNet-50 on ImageNet are \{72.51, 72.57, 72.64, 72.56, 72.65, 72.70, 72.58, 72.60, 72.56, 72.54\}
>     * We use global pruning and do not set hierarchical pruning rates. And the third last line in Algorithm 1 minimizes the reconstruction loss.
>
> 5. **Explanation about the state "the sparsity of different layers is inconsistent in a pruning group" (Q3).**
>     * It is our motivation for adopting sparse penalty. Channels with the same index in different layers in a group will be pruned together. However, these channels do not all have small weight norms. The combined importance of small-norm channels and large-norm channels may also be below the pruning threshold. And pruning them causes information loss. Therefore, we impose a sparsity penalty to make the pruned channels have consistent sparsity.
>
> [1] Pruning filters for efficient convnets. Li et al. ICLR 2017.
> [2] Neural Pruning via Growing Regularization. Wang et al. NeurIPS 2021.
> [3] DepGraph: Towards Any Structural Pruning. Fang et al. CVPR 2023.
> [4] DFPC: Data flow driven pruning of coupled channels without data. Narshana et al. ICLR 2023.

---

> > ### Author Response · Authors · 2023-11-21
> > **Additional Response to Reviewer 4NdE**
> >
> > Dear Reviewer 4NdE,
> >
> > We wanted to express our gratitude for reviewing our submission. Your feedback was very helpful, and we have done our best to address it in our rebuttal. As the second phase of the rebuttal process is coming to a close, we would be grateful if you could acknowledge receipt of our rebuttal and let us know if it addresses your comments. We would also be happy to engage in further discussions if needed. Thank you again for your expertise and assistance.

---

> > ### Comment · Reviewer_4NdE · 2023-11-23
> > **Thank you for your rebuttal**
> >
> > The authors have addressed most of my concerns. However, per the points raised by reviewers eiUM [regarding demonstrating performance of your finetuning framework (vs. standard finetuning) on different saliency scores] and 2uwp [regarding novelty and distinction of the submitted work from [a]], I will maintain my score.
> >
> > [a] MEST: Accurate and Fast Memory-Economic Sparse Training Framework on the Edge, Neurips 2021.

---

### Official Review · Reviewer_2uwp · 2023-10-31

**Soundness:** 2 fair
**Presentation:** 2 fair
**Contribution:** 2 fair
**Rating:** 5
**Confidence:** 3

**Summary:**

In this paper the authors propose a method to extend unstructured pruning approaches to induce structured sparsity. The method accomplishes this by adopting a 2 phase reconstruction strategy where the first step attempts to separate filters into 2 non-overlapping sets where 1 set contains the consolidated information of the original group of filters, with a large weight normalization, and another set that contains very little residual information, with a small weight normalization. After this decomposition a relatively straightforward second step prunes the newly formed filters with low weight normalization. The authors approach is predicated on the use of a small calibration dataset to perform the weight reconstruction during the first phase and another update during the pruning phase. The results provided demonstrate that the approach yields much better results compared to other competitive methods that are unable to scale favorably to large networks on large datasets. The results show much better performance on CIFAR10/100 and Imagenet datasets, in terms of model accuracy, while maintaining a structured pattern that may yield better performance compared to unstructured methods.

**Strengths:**

- The proposed method represents a natural extension of previously proposed methods to induce structured sparsity while maintaining accuracy.
- Compacting information across filters by considering the action of a group of filters prior to the pruning process seems to be a favorable direction for increasing sparsity.
- Minimizing the amount of data required for the training steps also makes the method attractive and interesting since the amount of data and the number of calibration steps are shown the be small.
- The strategy for transitioning the loss functions to account for the structural connectedness between filters and encourage sparsity is clever and naturally extends the work presented in previous papers.
- The authors proposed a balanced updating strategy to distribute the gradient updates across multiple layers.
- Results for applying the methods for sparsification on ResNet and VGG models demonstrate notable improvement with respect to a number of other competing implementations.

**Weaknesses:**

Major:
- While the connection between the input to equation (2) is the parameter set w the definition only uses the layer weights W and the accompanying activations X. Also the the indexing over i is described in the text but requires a bit of rereading for the reader to grasp what the authors are trying to convey with the equation. Overall the readability suffers and I would appreciate a bit more attention to the connectedness of the variables.
- Connected to equation (2), the authors mention "naive optimizing...induces imbalanced parameter updates...earlier layers being over-optimized". I'm not sure I understand what imbalanced updates and over-optimization actually mean. Empirically the authors attempt to support this statement using Figure 6 to illustrate that earlier layers are updated more aggressively than later layers so the proposed update attempts to smooth out these updates. When earlier layers are optimized too much does this negatively impact the reconstruction loss in equation (2) because the pruned output activations are not and should not be closely correlated with the unpruned activations?
- Similar notational issues exist in equation (4) as mentioned in equation (2).
- "...to mitigate the side effect of the sparse penalty..." this sentence could use more explanation to ensure the reader understands what the authors are referring to as the side effect of the penalty.
- Algorithm 1 needs to be updating to fix issues with the definition with respect to the defined inputs. Although the pretrained model is defined as an input it isn't used anywhere in the actual definition.
- Table 2 in section 4.2 seems to be missing some information to delineate the first set of columns denoting the top-1, RP, and RF metrics from the second set.
- PSP yields similar results to AdaPrune in terms of RP and RF factors in Table 2 but with much better accuracy. I think a natural question would be the effectiveness of the structural pruning compared to the application of the calibration updating steps. It's possible the increased accuracy is due to the post pruning calibration steps, do any of the other non-training methods perform any post pruning calibration steps?
- In Figure 5 the results for ResNet18 on CIFAR100 seem to be particularly bad for "w/o REC" compared to the other graphs even for a small number of pruned parameters. Is there a reason for this sharp drop-off?

Minor:
- The writing and presentation of the formulas during the main text could use more polishing before publication.
- Stylistically the font using for equation (1) and (2) is not consistent with the font used for W and X in the main text.

**Questions:**

I have combined my questions and suggestions as comments made in the weaknesses section.

---

> ### Author Response · Authors · 2023-11-14
> **Response to Reviewer 2uwp**
>
> Dear Reviewer 2uwp,
>
> We are grateful for your time to review our paper and that you find value in our work. We respond to your concerns below.
>
> 1. **Explanation about imbalanced updates and over-optimization.**
>     * The pruned model lacks the activation values of some channels, and we can only reconstruct the activation values of the remaining channels, which introduces error. We consider holistic layer reconstruction loss to reduce error accumulation. However, holistic reconstruction loss causes gradients of different layers to be unbalanced and causes the model to update incorrectly.
>     For example, the first layer is optimized by the reconstruction error of itself and all subsequent layers, but the last layer is optimized by the reconstruction error of itself only. So, we need to balance the updates of different layers.
>
> 2. **Explanation about settings and results in Table2**
>     * The results of AdaPrune and gAP are produced by our extended version of structured pruning. We use the same framework and importance evaluation, prune the same structure and yield RP and RF factors. The difference lies in the calibration process, our method is more efficient in recovering the accuracy. The other two non-retraining methods are designed for structured pruning, so we do not need to modify. Neuron Merging merge parameters of neurons and does not require data for calibration. However, it yields the worst accuracy. Reborn Filters concentrates information on newly generated filters before pruning. This process requires calibration data. It can fine-tune or do nothing after pruning.
>
> 3. **Explanation of the results for ResNet18 on CIFAR100 in Figure 5**
>     * CIFAR-100 is more difficult than CIFAR-10. ResNet-18 trained on CIFAR-100 contains less redundancy, and small amounts of pruning can result in significant accuracy reduction.
>
> Besides, we have revised unclear expressions in an updated version.

---

> > ### Author Response · Authors · 2023-11-21
> > **Additional Response to Reviewer 2uwp**
> >
> > Dear Reviewer 2uwp,
> >
> > Thank you for your valuable feedback on our submission. We have read your comments carefully and have addressed them in our rebuttal. As the second phase of the rebuttal process is ending soon, we would be grateful if you could acknowledge if our responses have addressed your comments. We would also be happy to engage in further discussions if needed. Thank you again for your time and consideration.

---

### Official Review · Reviewer_aCSe · 2023-11-01

**Soundness:** 2 fair
**Presentation:** 2 fair
**Contribution:** 2 fair
**Rating:** 5
**Confidence:** 4

**Summary:**

This paper proposed a post-training framework using structured pruning by accessing a small amount of unlabeled data.

**Strengths:**

1. The framework achieves fairly good results compared with baselines.
2. The paper is written clearly and easy to follow.

**Weaknesses:**

[1] The novelty of this paper is limited. The weight importance has been widely used in pruning, and the formulation used in this paper seems pretty similar to previous work [*] (in an unstructured form). It's not clear why the group importance works so well in this paper.

[2] Motivation. I'm not sure about the point of using unlabeled data setting. Since the unlabeled data are still from the original dataset (which violates the data privacy regulations) and a small amount of labeled calibration input data is already a realistic setting.

[3] It's unclear why the group importance and two phases layer reconstruction strategy work so well in this paper. I suggest the author provide a more comprehensive experiment report to help us understand that.

[*] MEST: Accurate and Fast Memory-Economic Sparse Training Framework on the Edge, Neurips 2021.

**Questions:**

Could authors provide a more comprehensive experiment report to help us understand the effectiveness of group importance and the two phases layer reconstruction strategy (compared with other baselines)?

---

> ### Author Response · Authors · 2023-11-14
> **Response to Reviewer aCSe**
>
> Dear Reviewer aCSe,
>
> We are grateful for your valuable feedback on our paper and that you find strength in our framework. We present our response to your comments below.
>
> 1. **Contributions of our paper (W1).**
>     * Our main contribution is the post-training structured pruning framework and two-stage reconstruction method. We chose the commonly used weight criteria and group importance as the implementation, but it is not the focus of our paper. The key to achieving excellent results is our reconstruction method rather than group importance. As evidence, we extended AdaPrune and gAP using the same group importance but their performance was inferior to our method.
>
> 2. **Motivation of our paper (W2).**
>     * **Privacy or commercial concerns:** While full business data is difficult to obtain, companies may agree to provide small samples. Moreover, our focus is on the amount of data, not whether it is labeled
>     * **Computational cost of retraining:** The high cost of retraining is also one of the obstacles to the practical application of pruning technology. This is also the motivation for our retraining-free framework.
>
> 3. **Could authors provide a more comprehensive experiment report (W3)?**
>     * Figure 3 provides intuition for why sparsity penalty works. It reduces the information loss caused by pruning. Moreover, Figure 5 provides ablation experiments to demonstrate the role of reconstruction and sparsity penalty.  Besides, Figure 6 shows that balanced updates are key to holistic reconstruction loss success. We think we have provided a more comprehensive experiment. We sincerely hope to receive your specific experimental suggestions. By the way, we provided more experiments in the appendix and code in the supplementary material, which may help you understand our approach.

---

> > ### Author Response · Authors · 2023-11-21
> > **Additional Response to Reviewer aCSe**
> >
> > Dear Reviewer aCSe,
> >
> > Thank you for your valuable feedback on our submission. We have read your comments carefully and have addressed them in our rebuttal. As the second phase of the rebuttal process is ending soon, we would be grateful if you could acknowledge if our responses have addressed your comments. We would also be happy to engage in further discussions if needed. Thank you again for your time and consideration.

---

### Official Review · Reviewer_eiUM · 2023-11-01

**Soundness:** 2 fair
**Presentation:** 3 good
**Contribution:** 1 poor
**Rating:** 5
**Confidence:** 4

**Summary:**

This paper introduces an approach for quickly pruning pre-trained models without retraining. This method focuses on structured pruning, grouping dependent structures across layers and marking less significant channels for elimination. A unique two-phase layer reconstruction strategy is applied, leveraging a minuscule amount of unlabeled data to regain any lost accuracy from pruning. The first phase emphasizes sparsity on less vital channels to retain information in the remaining components pre-pruning. The subsequent phase involves pruning and recalibrating the pruned model's layer output to mirror the original model.

**Strengths:**

This paper is well-written and easy to follow.
It improves group-based structured pruning techniques with holistic approach.

**Weaknesses:**

-	The performance is mediocre compared to the SOTA data-free algorithms, such as RED++.

-	Pruning coupled structures is proposed by DepGraph. The remaining contribution is not significant, especially considering the limited performance improvement,

-	Retraining-based techniques for performance comparison should be chosen among more recent ones, not the naïve L1-norm.

-	The experiments are limited in the diversity of datasets and models.

-	Claiming this study as "no retraining" seems exaggerated, especially compared to genuine data-free methods since this research employs some training data.

**Questions:**

-	Is it possible to precisely control FLOPs and parameters of the pruned model?

-	In Section 3.2.1, what causes Equation 2 to over-optimize the earlier layers?

-	Could you compare the accuracy between calibration and the fine-tuning approach of standard pruning under the same group-wise pruning ratio settings? It can show the effectiveness of the calibration method.

-	Does the L1-norm in Tables 1 and 2 use all the training data? In that case, the accuracy seems too low.

-	Why VGG-19 is used instead of VGG-16?

---

> ### Author Response · Authors · 2023-11-14
> **Response to Reviewer eiUM**
>
> Dear Reviewer eiUM,
>
> We are grateful for your valuable feedback on our paper and that you find value in our work. We present our response to your comments below.
> 1. **About baseline selection**
>     * Our paper focuses on pruning without retraining, so the pruning methods with the same settings is the focus of our comparison. We have compared rich and representative baselines. As for red++ ([4]), since it does not provide code and we failed to reproduce their results, this method has not been selected as a baseline.  In addition, in order to show the gap with the retraining method, we selected the simple $L_1$-norm based baseline in Table 1 and Table 2, which we retrained 100 epochs after pruning.
>     We have added the results of [1] and [2] in the revision.
>
> 2. **Contributions and significant performance improvement of our paper**
>     * Our main contribution is the post-training structured pruning framework and two-stage reconstruction method. Compared with methods that do not require retraining, our method has significant improvements, especially on the large-scale dataset ImageNet. Other reviewers have appreciated our performance improvements.
>
> 3. **Explanation for "no retraining"**
>     * No retraining means avoiding costly training processes after pruning rather than no data.  Our framework replaces tedious retraining with a simple and fast calibration process. Moreover, a small amount of calibration data is a realistic setting.
>
> 4. **Dataset and model setup**
>     * **Dataset**: CIFAR-10/100 and ImageNet datasets are the common settings for pruning papers.
>     * **Dataset**: We use the same settings as other papers for retraining-free pruning. And we use VGG-19 rather than VGG-16 for consistent settings. Besides, we provided results for more models in Appendix A.3.
>
> 5.  **Explanation about and over-optimization.**
>     * Holistic reconstruction loss causes gradients of different layers to be unbalanced. For example, the first layer is optimized by the reconstruction error of itself and all subsequent layers, but the last layer is optimized by the reconstruction error of itself only. So, we need to balance the updates of different layers.
>
> 6. **About pruning ratio settings.**
>     * Our implementation using global pruning rate does not allow precise control of model parameters and FLOPs. We have used the same group-wise pruning wherever possible. We extended AdaPrune and gAP to use group-wise pruning. However, Neuron Merging and Reborn Filters are difficult to be compatible with group pruning, so we used the original method.
>
> [1] Neural Pruning via Growing Regularization. Wang et al. NeurIPS 2021.
> [2] DepGraph: Towards Any Structural Pruning. Fang et al. CVPR 2023.
> [3] DFPC: Data flow driven pruning of coupled channels without data. Narshana et al. ICLR 2023.
> [4] Red++: Data-free pruning of deep neural networks via input splitting and output merging. Yvinec et al. TPAMI 2022.

---

> > ### Comment · Reviewer_eiUM · 2023-11-20
> >
> > I would like to thank the authors for the response and some of my concerns and questions are resolved.
> > Thus, I increase my score to "marginally below the acceptance threshold".

---

### Meta-Review · Area_Chair_D6G9 · 2023-12-11

**Metareview:**

Authors propose a method to do structured pruning by grouping parameters according to layer dependencies and imposing regularization to transfer "information" from less important parameters to parameters that are retained. Subsequently the pruning is executed and a small unlabeled dataset is used to recover the lost accuracy. The main concern raised in the reviews is about the novelty of the work over existing pruning papers where core ideas of the paper have already appeared. While the reviewers acknowledge the combination of these ideas in the paper is interesting but the empirical gains are deemed to be marginal over SOTA methods. Unfortunately these points together land the paper below the acceptance bar at ICLR.

**Justification For Why Not Higher Score:**

Concerns from the reviewers about novelty of the ideas over existing work and marginal benefits in terms of empirical gains.

**Justification For Why Not Lower Score:**

N/A

---

### Decision · Program_Chairs · 2024-01-16

Reject